# Self-medication practice and contributing factors among pregnant women

**Yirga Legesse Niriayo**[1]*, **Kadra Mohammed**[1], **Solomon Weldegebreal Asgedom**[1], **Gebre Teklemariam Demoz**[2], **Shishay Wahdey**[3], **Kidu Gidey**[1]

**1** Department of Clinical Pharmacy, School of Pharmacy, College of Health Sciences, Mekelle University, Mekelle, Tigray, Ethiopia, **2** Departments of Pharmacy, Clinical Pharmacy and Pharmacy Practice Unit, College of Health Sciences, Aksum University, Axum, Tigray, Ethiopia, **3** Department of Public Reproductive Health, School of Public Health, College of Health Sciences, Mekelle University, Mekelle, Tigray, Ethiopia

* yirga.legesse@mu.edu.et, yirga.pharma@gmail.com

## Abstract

### Background

The practice of self-medication during pregnancy is a global challenge that necessitates high attention as it poses a potential threat to the pregnant mother and fetus. However, little is known regarding self-medication practice and its contributors among pregnant women in our setting.

### Objective

The main aim of this study was to investigate the practice of self-medication and its contributing factors among pregnant women

### Methodology

A cross sectional study was conducted among pregnant women at antenatal care follow-up of Ayder comprehensive specialized hospital, Tigray, Ethiopia. Written informed consent was obtained from each participant before interview. Simple random sampling technique was employed to recruit participants in to the study. Data were collected by interviewing participants using the structured questionnaire. Binary logistic regressions analysis was performed to determine the contributing factors of self-medication practice during pregnancy. A p value of less than 0.05 was considered as significant.

### Results

A total of 250 pregnant women were included in the study. Of the total, 40.8% practiced self-medication during the current pregnancy. Morning sickness (39.2%), headache (34.3%), and upper respiratory tract infections (29.4%) were the leading indications for self-medication. According to participant report, ease of access to medicines (25.5%), feelings that the disease is minor (21.6%) and timesaving (19.6%) were the most commonly reported reasons for self-medication practice. Absence of health insurance (AOR: 2.75, 95%CI: 1.29–5.89) and being on first trimester of pregnancy (AOR: 2.44, 95%CI: 1.02–5.86) were significant contributors of self-medication practice among pregnant women.

**Data Availability Statement:** All relevant data are within the manuscript and its Supporting Information files.

**Funding:** The author(s) received no specific funding for this work.

**Competing interests:** The authors have declared that no competing interests exist.

## Conclusion

In our study, high prevalence of self-medication was reported among pregnant women. Self-medication practice during pregnancy was higher among pregnant women on first trimester and those who were not having health insurance. Therefore, intervention programs should be designed to minimize the practice of self-medication during pregnancy.

## Introduction

According to the World Health Organization (WHO), self-medication is defined as the act of using medications by patients or individuals to treat self-diagnosed disorders or symptoms on their own initiative without getting advice from health care provider [1–3]. Self- medication includes use of over counter drugs available without a physician's prescription, irregular use of prescribed medicines, use of leftover drugs from previous prescriptions, and use of herbal and traditional medicines [4,5]. Owing to the limited access to health care service and scarcity of resources, the practice of self-medication is common in developing countries including Ethiopia [6,7]. According to the meta-analysis reported in 2018, the prevalence of self-medication in Ethiopia ranged from 12.5%-78.1% with an average of 44% [8].

The practice of self-medication during pregnancy has been increasing worldwide, particularly in developing countries owing to the poor health care system [9–11]. Likewise, the practice of self-medication during pregnancy is increasing in Ethiopia with a reported prevalence ranged from 15.5%-70% [8,12,13].

In clinical practice, self-medication during pregnancy still remained a significant medical challenge [14,15]. Although drugs are frequently used during pregnancy in clinical practice, their safety has not been well-established as pregnant women are often excluded from clinical trials due to the fear of harm on the mother or the developing fetus [15,16]. Therefore, unless absolutely necessary, drugs should be avoided during pregnancy [17]. Despite this fact, there is high level of self-medication use during pregnancy [18]. Self-medication can affect both the fetus and the mother and it could cause detrimental effects on the fetus including malformation/teratogenicity, fetal toxicity, low birth weight, premature birth, respiratory problems as well as death [15,18,19]. It has been reported that at least 10% of birth defects are resulted from the exposure of pregnant women to drugs [15].

The prevalence of self-medication varies across different communities and it could be affected by several factors including lack of access to healthcare service, unregulated distribution of medicines, patients' attitudes toward healthcare providers, socio-economic factors, long waiting times, cost of the drugs, educational level, age, income, education level, satisfaction, and belief of people's toward medication and disease [9,12,20–22].

Despite the potential harmful effect of self-mediation practice during pregnancy [15,16], there is little awareness about the impact of self-medication practice among pregnant mothers in developing countries including Ethiopia [7,10,23,24]. Hence, evaluation of self-medication practice and its contributors will provide information for health policy makers and relevant stakeholders to develop strategies to prevent the risks associated with self-medication practice during pregnancy. In our setting, though there are some studies on self-medication in general population [7,8], studies regarding self-medication practices during pregnancy are scanty. Our study, therefore, investigated the practice of self-medication and its contributing factors during pregnancy.

## Methodology

**Study design and setting.** An institutional based cross-sectional study was conducted from January to April 2019 at antenatal care follow up of Ayder comprehensive specialized hospital (ACSH), Tigray region, Ethiopia. ACSH is the largest public hospital in Tigray region whiles it the second in Ethiopia next to black lion hospital. It provides service for about 10 million people in the catchment area.

**Study participants and data collection procedure.** All pregnant mothers who had antenatal care (ANC) follow up in ACSH were the source population. Pregnant women at any gestational age who came for ANC to ACSH hospital during the study period were included in this study. Pregnant women who are critically ill to give response, unable to hear/communicate and those who are unwilling to give consent were excluded from the study.

The sample size for this study was determined using the single population proportion formula for the prevalence of self-medication practice. Accordingly a sample of 262 participants was calculated assuming 26.6% prevalence of self-medication practice during pregnancy according to a study conducted in Addis Abeba [24], 95% confidence level, 5% margin of error, and 10% contingency for nonresponse rate. From 262 participants approached, 12 patients were excluded from the study due to unable to hear [1], critically ill to give response [4], and unwillingness to give consent [7]. The participants were recruited into the study during their appointment for ANC using simple random sampling technique.

We collected the data using an interviewer administered structured questionnaire (**S1 Table**). The questionnaire was developed based on previous studies [9,13,14] and it was amended to fit the current study. The questionnaire was translated to local language (Tigrigna), and then back translated to English to ensure consistency of meaning. Pre-test was carried out on 5% of the sample before the commencement of the actual data collection and slight amendments were made on the questionnaire based on the findings. The questionnaire involves data related to socio-demographics, obstetrics factor, and self-medication practices. Fifth year clerkship pharmacy students were employed to collect the data for this study and they were given training and orientation.

**Statistical analysis.** We analyzed the data using the Statistical Package for the Social Science (SPSS version 24.0) (**S2 Table**). Descriptive statics was computed as frequency, mean and standard deviation (SD). Multicollinearity was checked to test correlation among predictor variables using variance inflation factor (VIF) and none was collinear. The association of each independent variable with self-medication practice was determined using univariable logistic regression analysis. Furthermore, the variables with p value <0.25 in univariable analysis were re-analyzed using multivariable binary logistic regression model to identify the independent predictors of self-medication practice during pregnancy. A p value of <0.05 was considered statistically significant in all analyses.

**Ethics.** This study was approved by ethics review committee of school of pharmacy, College of Health Sciences, Mekelle University. Each study participant was well informed about the objective of the study. After getting permission from the ACSH hospital, written informed consent was obtained from all participants. Confidentiality was assured for all the information provided. All the methods were performed in accordance with approved institutional guidelines.

## Result

### Socio-demographic characteristics

A total of 250 participants were included in this study. The mean (±SD) age was 26.9±5.42. Most (83.6%) of the participants have attended primary school and above. Nearly half (46.4%) were

**Table 1. Socio-demographic characteristics of participants (n = 250).**

| Category | Frequency (%) |
|---|---|
| **Age** | |
| ≤18 | 2(0.8) |
| 19–25 | 115(46) |
| 26–30 | 83(33) |
| 31–35 | 34(13.6) |
| >35 | 16(6.4) |
| **Residence** | |
| Urban | 183(73.2) |
| Rural | 67(26.8) |
| **Educational status** | |
| No formal education | 36(14) |
| Primary | 41(16.4) |
| Secondary | 108(43.2) |
| Higher education | 65(26) |
| **Occupation** | |
| Civil servant | 59(23.6) |
| Merchant | 50(20) |
| Housewife | 116(46.4) |
| Others | 25(10) |
| **Income** | |
| <5000 | 126(50.4) |
| > = 5000 | 124(49.6) |
| **Chronic illness** | |
| Yes | 11(4.4) |
| No | 239(95.6) |
| **Alcohol** | |
| No | 207(82.8) |
| Yes | 43(17.2) |
| **Health insurance** | |
| Yes | 53(21.2) |
| No | 197(78.8) |

housewives. Majority of the participants were from urban. The mean income was 4543.42± 3436.23 Ethiopian Birr. Alcohol consumption was reported in 17.2% of the participants (Table 1).

## Obstetric factors

Nearly half (44.8%) of the participants were in the first trimester of their pregnancy and majority (66%) were multigravidas. About two-thirds (62.8%) of participants had ANC follow-up in their previous pregnancy and 16.8% experienced complications related to previous pregnancy (Table 2).

## Self-medication practice during pregnancy

Of the total, 40.8% practiced self-medication during the current pregnancy while one-fourth (25.2%) of participants had previous self-medication experience. Among those who used self-medication (102), 43(42.2%) medicated themselves with modern medicine and 41(40.2%) used traditional medicine while 18(17.6%) used both modern medicine and traditional

**Table 2. Obstetrics characteristics of study participants, 2019(n = 250).**

| Category | Frequency (%) |
|---|---|
| **Gestational period** | |
| First trimester | 112(44.8) |
| Second trimester | 101(40.4) |
| Third trimester | 37(14.8) |
| **Gravidity** | |
| Primeravida | 85(34) |
| Mutigravida | 165(66%) |
| **Previous ANC follow up** | |
| Yes | 157(62.8) |
| No | 93(37.2) |
| **Place of delivery of last baby** | |
| Home | 8(4.8) |
| Health institution | 157(93.2) |
| **Previous still birth** | |
| No | 227(90.8) |
| Yes | 23(9.2) |
| **Previous pregnancy related complications** | |
| Yes | 42(16.8) |
| No | 208(83.2) |

medicine. The participants mentioned deferent reasons for self-medication. Among those, the major reasons for self-medication were easily accessing medicines (25.5%), feeling that the disease is minor (21.6%), and timesaving (19.6%). Morning sickness (39.2%), headache (34.3), and upper respiratory tract infections (29.4%) were the most common indications for self-medication (Table 3).

## Factors associated with self-medication practice during pregnancy

Independent variables including age, residence, educational status, occupation, income, presence of chronic illness, alcohol, health insurance, gestational period, gravidity, still birth, delivery place of last baby, prior pregnancy related complications, and prior ANC follow-up were analyzed using univariable logistic regression analysis to assess their association with self-medication practice.

Accordingly, gestational age (COR:2.90,95%CI:1.25–6.70), health insurance (COR:3.06,95% CI:1.63–5.74), gravidity (COR:1.79, 95%CI:1.03–3.10) were significantly associated with self-medication practice in univariate analysis. Moreover, variables with P<0.25 in the univariable analyses including residence, educational status, chronic illness, health insurance, gestational age, gravidity, and previous pregnancy related complications were re-analyzed using multivariable logistic regression model. The whole model containing all predictors was statistically significant (Chi-square = 27.676, df = 10, P = 0.002). In multivariable logistic regression analysis, participants without health insurance (AOR: 2.75, 95%CI: 1.29–5.89) and participants on first trimester (AOR: 2.44, 95%CI: 1.02–5.86) were more likely to practice self-medication compared to those with health insurance and on third trimester, respectively (Table 4).

## Discussion

The practice of self-medication is a global challenge that necessitates high attention because it poses a potential threat to the pregnant mother and fetus [14,25]. In developing countries

**Table 3. Self-medication practice during pregnancy, 2019(n = 250).**

| Characteristics | Frequency (%) |
|---|---|
| **Self-medication** | |
| Yes | 102(40.8) |
| No | 148(59.2) |
| **History of previous self-medication** | |
| Yes | 63(25.2) |
| No | 187(74.8) |
| **Type of medicine used** | |
| Modern only | 43(42.2) |
| TDM only | 36(18.5) |
| Both modern and TDM | 18(17.6) |
| **Reason for self-medication** | |
| Easily accessing medicines | 26(25.5) |
| Disease not serious | 22(21.6) |
| Timesaving | 20(19.6) |
| Poor health service provision | 15(14.7) |
| Cost saving | 13(12.7) |
| Lack of trust on prescribers | 6(5.9) |
| **Common Indications for self-medication** | |
| Morning sickness | 40(39.2) |
| Headache | 35(34.3) |
| Upper respiratory tract infections | 30(29.4) |
| Dyspepsia | 22(21.6) |
| Urinary tract infections | 20(19.6) |
| Cough and cold | 16(15.7) |
| Diarrhea | 14(13.7) |
| Allergic rhinitis | 8(7.8) |
| **Source of modern medication for self-medication** | |
| Pharmacies/drug stores | 60(24) |
| Leftover medicine | 32(12.8) |
| Sharing with family, friends or neighbors | 10(4) |

including Ethiopia, clinicians may not be aware of the actual burden of self-medication and its contributing factors during pregnancy. Thus, conducting such kind of study will help them to design and develop strategies to prevent/minimize self-medication practice during pregnancy. Our study, therefore, investigated the practice of self-medication and its contributing factors among pregnant women. The current study revealed that a significant proportion of pregnant women practiced self-medication either with modern and/or herbal medicine without consulting healthcare professionals.

Despite the potential harmful effect of self-mediation during pregnancy [15,16], nearly half (40.8%) of pregnant women practiced self-medication during their current pregnancy. This result is comparable with the findings reported from Tanzania [14] and Iran [22]. In contrast, higher findings were reported from previous studies conducted in Democratic Republic of Congo [26], Nigeria [27] and Ethiopia [12]. On the other hand, our finding is higher compared to the findings reported from Addis Abeba [24] and Netherland [28]. These variations could be attributed to the differences in level of awareness about risks of self-medication in pregnancy, population demographics, access to healthcare service, and restriction policies of dispensing practices.

**Table 4. Factors associated with self-medication practice during pregnancy (n = 250).**

| Characteristics | Self-medication | | COR (95% CI) | p-value | AOR (95% CI) | p-value |
|---|---|---|---|---|---|---|
| | No, n (%) | Yes, n(%) | | | | |
| **Residence** | | | | | | |
| Urban | 113(76.4) | 70(68.6) | 1 | | 1 | |
| Rural | 35(23.6) | 32(31.4) | 1.48(0.84–2.60) | 0.177 | 0.80(0.37–1.74) | 0.577 |
| **Educational status** | | | | | | |
| No formal education | 18(12.2) | 18(17.) | 1.41(0.62–3.19) | | 0.82(0.29–2.29) | 0.702 |
| Primary | 17(11.5) | 24(23.5) | 1.99(0.90–4.30) | 068 | 1.30(0.52–3.21) | 0.574 |
| Secondary | 75(50.7) | 33(32.4) | 0.62(0.33–1.17) | | 0.60(0.30–1.15) | 0.123 |
| College and above | 38(25.7) | 27(26.5) | 1 | | 1 | |
| **Chronic illness** | | | | | | |
| No | 144(97.3) | 95(93.1) | 1 | | 1 | |
| Yes | 4(2.7) | 7(6.9) | 2.653(0.756–9.310) | 0.128 | 1.228(0.237–6.36) | 0.807 |
| **Health insurance** | | | | | | |
| Yes | 20(13.5) | 33(32.4) | 1 | | 1 | |
| No | 128(86.5) | 69(67.6) | 3.061(1.634–5.735) | <0.001 | 2.75(1.29–5.89) | 0.009 |
| **Gravidity** | 58(39.2) | 27(26.5) | 1 | | 1 | |
| Primigravida | 90(60.8) | 75(73.5) | 1.228(0.64- 2.354) | 0.537 | 0.63(0.34–1.18) | 0.152 |
| Multigravida | 58(39.2) | 27(26.5) | 1 | | 1 | |
| **Previous pregnancy related complication** | | | | | | |
| No | 128(86.5) | 80(78.4) | 1 | | 1 | |
| Yes | 20(13.5) | 22(21.6) | 2.669(1.202–5.926) | 0.016 | 1.20(0.56–2.61) | 0.637 |
| **Gestational age** | | | | | | |
| First trimester | 58(39.2) | 54(52.9) | 2.99(1.25–6.69) | 0.036 | 2.44(1.02–5.9) | 0.045 |
| Second trimester | 62(41.9) | 39(38.2) | 1.96(0.84–4.58) | | 1.87(0.77–4.54) | 0.166 |
| Third trimester | 28(18.9) | 9(8.8) | 1 | | 1 | |

Morning sickness, headache, and upper respiratory tract infections were the leading indications for self-medication in this study. In Tanzanian study [14], malaria, morning sickness, and headache were the leading illness that led to self-medication. Unlike our study, malaria was the most common indication for self-medication in Tanzanian study [14]. This could be due to the less prevalence of malaria in our study setting.

In the present study, the most commonly reported reasons for self-medication practice during pregnancy were ease of access to medicines, feeling that disease is minor and prolonged waiting time. In agreement with our study, similar finding were reported from previous studies conducted in Addis Ababa [24] and democratic republic of Congo [26]. The ease of access to medications without prescription could be attributed to the poor drug regulatory system and weak enforcement on restricting prescription drugs sale without prescription as well as nonprescription drugs sale to pregnant women. Moreover, it could be augmented due to the lack of attention and priorities of health policy makers and other stakeholders on the burden of self-medication risks [24]. Therefore, necessary measures should be taken to strengthen regulatory system and enforce regulations so as to reduce the practice of self-medication during pregnancy.

Our study revealed that pregnant women without health insurance were more prone to self-medication practice which is consistent with a finding reported from Iran [22]. The possible explanation is that those who have health insurance are more likely to visit health institution and see a doctor. Thus, they are more likely to get prescribed medication as the cost of their visit and medication can be compensated by the insurance.

Pregnant women on first trimester were more likely to practice self-medication compared to those on third trimester. This finding is supported by a study conducted in Tanzania [14]. The possible justification for higher self-medication practice during first trimester of pregnancy could be due to the more frequent occurrence of symptoms and/or illnesses including morning sickness, headache, and fever in the first trimester than other trimester during pregnancy. More importantly, this finding is worrisome as drug exposure in this period is more likely to cause congenital abnormalities [29–31]. Therefore, more emphasis should be given to the use of medication during first trimester of pregnancy.

Finally, our study had some limitations. Our study was a cross–sectional suggesting that it cannot provide adequate evidence of causality regarding self-medication and its contributing factors. During interview, pregnant women were expected to recall information from their past experience; therefore, recall bias might affect the study findings. The findings of our study could be affected by the difference in population demographics, healthcare system and regulations and knowledge of the participants. Therefore, it should be extrapolated to other countries with caution.

## Conclusion

In our study, a high prevalence of self-medication was reported among pregnant women. Self-medication practice during pregnancy was higher among pregnant women on first trimester and those who were not having health insurance. Therefore, healthcare providers should provide more emphasis to the risky participants and implementation of national health insurance needs to be encouraged. Moreover, intervention programs should be designed to minimize the practice of self-medication during pregnancy.

## Supporting information

**S1 Table. Data collection instrument.**
(DOC)

**S2 Table. Dataset.**
(SAV)

## Acknowledgments

We would like to acknowledge the data collectors and the hospital staff for their genuine cooperation. Our gratefulness goes to the pregnant women for their eager involvement in the study.

## Author Contributions

**Conceptualization:** Yirga Legesse Niriayo, Kadra Mohammed, Gebre Teklemariam Demoz.

**Data curation:** Yirga Legesse Niriayo, Kidu Gidey.

**Formal analysis:** Yirga Legesse Niriayo, Kadra Mohammed, Solomon Weldegebreal Asgedom, Kidu Gidey.

**Investigation:** Yirga Legesse Niriayo, Kadra Mohammed, Kidu Gidey.

**Methodology:** Yirga Legesse Niriayo, Kadra Mohammed, Solomon Weldegebreal Asgedom, Gebre Teklemariam Demoz, Shishay Wahdey.

**Project administration:** Yirga Legesse Niriayo, Kadra Mohammed.

**Resources:** Yirga Legesse Niriayo.

**Software:** Yirga Legesse Niriayo.

**Supervision:** Yirga Legesse Niriayo, Shishay Wahdey, Kidu Gidey.

**Validation:** Yirga Legesse Niriayo, Gebre Teklemariam Demoz.

**Visualization:** Yirga Legesse Niriayo, Solomon Weldegebreal Asgedom.

**Writing – original draft:** Yirga Legesse Niriayo, Kadra Mohammed.

**Writing – review & editing:** Yirga Legesse Niriayo, Kadra Mohammed, Solomon Weldegebreal Asgedom, Gebre Teklemariam Demoz, Shishay Wahdey, Kidu Gidey.

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
