## [Decision Letter · Decision Letter 0]

4 Dec 2020

PONE-D-20-32462

Self-medication practice and contributing factors among pregnant women

PLOS ONE

Dear Dr. Niriayo,

Thank you for submitting your manuscript to PLOS ONE. After careful consideration, we feel that it has merit but does not fully meet PLOS ONE’s publication criteria as it currently stands. Therefore, we invite you to submit a revised version of the manuscript that addresses the points raised during the review process.

We look forward to receiving your revised manuscript.

Kind regards,

Denis Bourgeois

Academic Editor

PLOS ONE

Journal Requirements:

2.  For more information on PLOS ONE's expectations for statistical reporting, please see https://journals.plos.org/plosone/s/submission-guidelines.#loc-statistical-reporting. Please update your Methods and Results sections accordingly.

3. Please include additional information regarding the survey or questionnaire used in the study and ensure that you have provided sufficient details that others could replicate the analyses. For instance, if you developed a questionnaire as part of this study and it is not under a copyright more restrictive than CC-BY, please include a copy, in both the original language and English, as Supporting Information.  If the original language is written in non-Latin characters, for example Amharic, Chinese, or Korean, please use a file format that ensures these characters are visible.

4.  We noticed you have some minor occurrence of overlapping text with the following previous publication(s), which needs to be addressed:

- https://tropmedhealth.biomedcentral.com/articles/10.1186/s41182-018-0091-z

In your revision ensure you cite all your sources (including your own works), and quote or rephrase any duplicated text outside the methods section. Further consideration is dependent on these concerns being addressed.

Reviewers' comments:

Reviewer's Responses to Questions

**Comments to the Author**

1. Is the manuscript technically sound, and do the data support the conclusions?

Reviewer #1: Yes

Reviewer #2: Yes

2. Has the statistical analysis been performed appropriately and rigorously? 

Reviewer #1: Yes

Reviewer #2: Yes

3. Have the authors made all data underlying the findings in their manuscript fully available?

Reviewer #1: Yes

Reviewer #2: Yes

4. Is the manuscript presented in an intelligible fashion and written in standard English?

Reviewer #1: Yes

Reviewer #2: Yes

5. Review Comments to the Author

Reviewer #1: Overall, the text is well organized and structured according to IMRED. The writing style is pleasant. However, the presentation of tables does not meet current standards for presentation of a table. It will remove the left and right borders and remove the intermediate lines only the bottom line and the variable banner (the 2 top lines).

There are also shortcomings in the presentation of the references, in particular reference 17 to be corrected.

The study population is well specified. Sample size and selection methods are also described.The ethical and regulatory aspects have been taken into account.

The analyses are in agreement with the study scheme. Statistical analyses are carried out to high technical standards and are described in sufficient detail. However, we should have presented the results of the univariate analysis with the overall p value for a variable instead of the p value per variable category; especially for variables with more than 2 modalities. In the presentation of table 4, for the reference modality, the p value must not appear (in yellow in the table). The presentation of Table 4 needs to be improved. Make a clear distinction between the variables that were used for the univariate analysis and those that were retained for the multivariate analysis.

The results meet the targeted objectives. The limits and biases as well as the impact of its biases are well mentioned.

The manuscript as a whole is well organized and clearly written enough to be accessible even to non-specialists. The few shortcomings noted have no major impact on the study, which remains solid but which requires some corrections.

Reviewer #2: This is a very interesting study : it could help to support and develop information and national and international prevention programs for pregnant women about the dangers of automédication for themselves and their fetus.

6. PLOS authors have the option to publish the peer review history of their article (what does this mean?). If published, this will include your full peer review and any attached files.

Reviewer #1: **Yes: **Jocelyne V. W. GARE

Reviewer #2: No

---

## [Author Response · Author response to Decision Letter 0]

29 Apr 2021

Manuscript number: PONE-D-20-32462

Title: “Self-medication practice and contributing factors among pregnant women” 

Authors: Yirga Legesse Niriayo*1, Kadra Mohammed1, Solomon Weldegebreal Asgedom1, Gebre Teklemariam Demoz2, Shishay Wahdey3 Kidu Gidey 1,

 Authors’ response to academic editor’s and reviewers’ comments 

We thank the academic editor and the reviewer for reviewing our manuscript. We greatly appreciate the academic editor and reviewers for their constructive comments and suggestions on our submitted manuscript. We have modified our manuscript based on the editor and reviewer comments and suggestions. We offer below responses to each of the points raised by the academic editor and reviewers. We have also attached the modified manuscript with track changes and without track change based on the editor’s and reviewers’ comments and suggestions with our resubmission. Please note that all page and line numbers we have mentioned below refer to the resubmitted manuscript with track changes. 

Response to academic editor comments:

Journal requirements 

Response: We have ensured that all style requirements are addressed. 

2. For more information on PLOS ONE's expectations for statistical reporting, please see https://journals.plos.org/plosone/s/submission-guidelines.#loc-statistical-reporting. Please update your Methods and Results sections accordingly

Response: We thank you. We have done so with our resubmission. 

3. Please include additional information regarding the survey or questionnaire used in the study and ensure that you have provided sufficient details that others could replicate the analyses. For instance, if you developed a questionnaire as part of this study and it is not under a copyright more restrictive than CC-BY, please include a copy, in both the original language and English, as Supporting Information. If the original language is written in non-Latin characters, for example Amharic, Chinese, or Korean, please use a file format that ensures these characters are visible. 

We have made all data, including data set and data collection tool fully available as supporting information.

Response: We have included the data collection tool used in this study that contains both English and Tigrigna version as supporting information. 

4. We noticed you have some minor occurrence of overlapping text with the following previous publication(s), which needs to be addressed:

- https://tropmedhealth.biomedcentral.com/articles/10.1186/s41182-018-0091-z

In your revision ensure you cite all your sources (including your own works), and quote or rephrase any duplicated text outside the methods section. Further consideration is dependent on these concerns being addressed.

Response: We have paraphrased some of the overlapping texts. Actually, the overlaps are occurred just by chance. Moreover, we have cited all utilized resources. 

. 

Response to Reviewer comments

1. Overall, the text is well organized and structured according to IMRED. The writing style is pleasant. However, the presentation of tables does not meet current standards for presentation of a table. It will remove the left and right borders and remove the intermediate lines only the bottom line and the variable banner (the 2 top lines

Response: We appreciate the reviewer comments. We have done so with our resumption. Please see the highlighted tables (Table 1, Table 2, Table 3, and Table 4) 

2. There are also shortcomings in the presentation of the references, in particular reference 17 to be corrected.

Response: We thank the reviewer’s comments and suggestions. We have corrected them with our resubmission. Please see the highlighted references with yellow colour. 

3. Statistical analyses are carried out to high technical standards and are described in sufficient detail. However, we should have presented the results of the univariate analysis with the overall p value for a variable instead of the p value per variable category; especially for variables with more than 2 modalities. 

Response: Thank you for your suggestions. Actually, both approaches (overall p value for a variable and p value per variable category) are possible. We have modified it according to your suggestions in our resubmission. Please see the highlights on table 4. 

4. In the presentation of table 4, for the reference modality, the p value must not appear (in yellow in the table). The presentation of Table 4 needs to be improved

Response: We appreciate the reviewer comments. We have done so with our resubmission. Please see the highlights on table 4. 

5. Make a clear distinction between the variables that were used for the univariate analysis and those that were retained for the multivariate analysis.

Response: We have done so with our re-submission. Please see the highlights on page 10 lines 209-221.

---

## [Editor Report · Decision Letter 1]

3 May 2021

Self-medication practice and contributing factors among pregnant women

PONE-D-20-32462R1

Dear Dr. Niriayo,

We’re pleased to inform you that your manuscript has been judged scientifically suitable for publication and will be formally accepted for publication once it meets all outstanding technical requirements.

Kind regards,

Denis Bourgeois

Academic Editor

PLOS ONE
---

## [Editor Report · Acceptance letter]

5 May 2021

PONE-D-20-32462R1 

Self-medication practice and contributing factors among pregnant women 

Dear Dr. Niriayo:

I'm pleased to inform you that your manuscript has been deemed suitable for publication in PLOS ONE. Congratulations! Your manuscript is now with our production department. 

Kind regards, 

on behalf of

Professor Denis Bourgeois 

Academic Editor

PLOS ONE